# Symbol as Points: Panoptic Symbol Spotting via Point-based Representation

**Wenlong Liu[1], Tianyu Yang[1], Yuhan Wang[2], Qizhi Yu[2] , Lei Zhang[1]**
[1]International Digital Economy Academy (IDEA)     [2]Vanyi Tech

## Abstract

This work studies the problem of panoptic symbol spotting, which is to spot and parse both countable object instances (windows, doors, tables, etc.) and uncountable stuff (wall, railing, etc.) from computer-aided design (CAD) drawings. Existing methods typically involve either rasterizing the vector graphics into images and using image-based methods for symbol spotting, or directly building graphs and using graph neural networks for symbol recognition. In this paper, we take a different approach, which treats graphic primitives as a set of 2D points that are locally connected and use point cloud segmentation methods to tackle it. Specifically, we utilize a point transformer to extract the primitive features and append a mask2former-like spotting head to predict the final output. To better use the local connection information of primitives and enhance their discriminability, we further propose the attention with connection module (ACM) and contrastive connection learning scheme (CCL). Finally, we propose a KNN interpolation mechanism for the mask attention module of the spotting head to better handle primitive mask downsampling, which is primitive-level in contrast to pixel-level for the image. Our approach, named SymPoint, is simple yet effective, outperforming recent state-of-the-art method GAT-CADNet by an absolute increase of 9.6% PQ and 10.4% RQ on the FloorPlanCAD dataset. The source code and models will be available at https://github.com/nicehuster/SymPoint.

## 1 Introduction

Vector graphics (VG), renowned for their ability to be scaled arbitrarily without succumbing to issues like blurring or aliasing of details, have become a staple in industrial designs. This includes their prevalent use in graphic designs(Reddy et al., 2021), 2D interfaces(Carlier et al., 2020), and Computer-aided design (CAD)(Fan et al., 2021). Specifically, CAD drawings, consisting of geometric primitives(e.g., arc, circle, polyline, etc.), have established themselves as the preferred data representation method in the realms of interior design, indoor construction, and property development, promoting a higher standard of precision and innovation in these fields.

Symbol spotting (Rezvanifar et al., 2019; 2020; Fan et al., 2021; 2022; Zheng et al., 2022) refers to spotting and recognizing symbols from CAD drawings, which serves as a foundational task for reviewing the error of design drawing and 3D building information modeling (BIM). Spotting each symbol, a grouping of graphical primitives, within a CAD drawing poses a significant challenge due to the existence of obstacles such as occlusion, clustering, variations in appearances, and a significant imbalance in the distribution of different categories. Traditional symbol spotting usually deals with instance symbols representing countable things (Rezvanifar et al., 2019), like table, sofa, and bed. Fan et al. (2021) further extend it to *panoptic symbol spotting* which performs both the spotting of countable instances (e.g., a single door, a window, a table, etc.) and the recognition of uncountable stuff (e.g., wall, railing, etc.).

Typical approaches (Fan et al., 2021; 2022) addressing the panoptic symbol spotting task involve first converting CAD drawings to raster graphics(RG) and then processing it with

powerful image-based detection or segmentation methods (Ren et al., 2015; Sun et al., 2019). Another line of previous works (Jiang et al., 2021; Zheng et al., 2022; Yang et al., 2023) abandons the raster procedure and directly processes vector graphics for recognition with graph convolutions networks. Instead of rastering CAD drawings to images or modeling the graphical primitives with GCN/GAT, which can be computationally expensive, especially for large CAD graphs, we propose a new paradigm that has the potential to shed novel insight rather than merely delivering incremental advancements in performance.

Upon analyzing the data characteristics of CAD drawings, we can find that CAD drawing has three main properties: 1). *irregularity and disorderliness*. Unlike regular pixel arrays in raster graphics/images, CAD drawing consists of geometric primitives(e.g., arc, circle, polyline, etc.) without specific order. 2). *local interaction among graphical primitives.* Each graphical primitive is not isolated but locally connected with neighboring primitives, forming a symbol. 3). *invariance under transformations.* Each symbol is invariant to certain transformations. For example, rotating and translating symbols do not modify the symbol's category. These properties are almost identical to point clouds. *Hence, we treat CAD drawing as sets of points (graphical primitives) and utilize methodologies from point cloud analysis (Qi et al., 2017a;b; Zhao et al., 2021) for symbol spotting.*

In this work, we first consider each graphic primitive as an 8-dimensional data point with the information of position and primitive's properties (type, length, etc.). We then utilize methodologies from point cloud analysis for graphic primitive representation learning. Different from point clouds, these graphical primitives are locally connected. We therefore propose contrastive connectivity learning mechanism to utilize those local connections. Finally, we borrow the idea of Mask2Former(Cheng et al., 2021; 2022) and construct a masked-attention transformer decoder to perform the panoptic symbol spotting task. Besides, rather than using bilinear interpolation for mask attention downsampling as in (Cheng et al., 2022), which could cause information loss due to the sparsity of graphical primitives, we propose KNN interpolation, which fuses the nearest neighboring primitives, for mask attention downsampling. We conduct extensive experiments on the FloorPlanCAD dataset and our SymPoint achieves 83.3% PQ and 91.1% RQ under the panoptic symbol spotting setting, which outperforms the recent state-of-the-art method GAT-CADNet (Zheng et al., 2022) with a large margin.

## 2 RELATED WORK

**Vector Graphics Recognition**  Vector graphics are widely used in 2D CAD designs, urban designs, graphic designs, and circuit designs, to facilitate resolution-free precision geometric modeling. Considering their wide applications and great importance, many works are devoted to recognition tasks on vector graphics. Jiang et al. (2021) explores vectorized object detection and achieves a superior accuracy to detection methods (Bochkovskiy et al., 2020; Lin et al., 2017) working on raster graphics while enjoying faster inference time and less training parameters. Shi et al. (2022) propose a unified vector graphics recognition framework that leverages the merits of both vector graphics and raster graphics.

**Panoptic Symbol Spotting**  Traditional symbol spotting usually deals with instance symbols representing countable things (Rezvanifar et al., 2019), like table, sofa, and bed. Following the idea in (Kirillov et al., 2019), Fan et al. (2021) extended the definition by recognizing semantic of uncountable stuff, and named it panoptic symbol spotting. Therefore, all components in a CAD drawing are covered in one task altogether. For example, the wall represented by a group of parallel lines was properly handled by (Fan et al., 2021), which however was treated as background by (Jiang et al., 2021; Shi et al., 2022; Nguyen et al., 2009) in Vector graphics recognition. Meanwhile, the first large-scale real-world FloorPlanCAD dataset in the form of vector graphics was published by (Fan et al., 2021). Fan et al. (2022) propose CADTransformer, which modifies existing vision transformer (ViT) backbones for the panoptic symbol spotting task. Zheng et al. (2022) propose GAT-CADNet, which formulates the instance symbol spotting task as a subgraph detection problem and solves it by predicting the adjacency matrix.

**Point Cloud Segmentation**  Point cloud segmentation aims to map the points into multiple homogeneous groups. Unlike 2D images, which are characterized by regularly

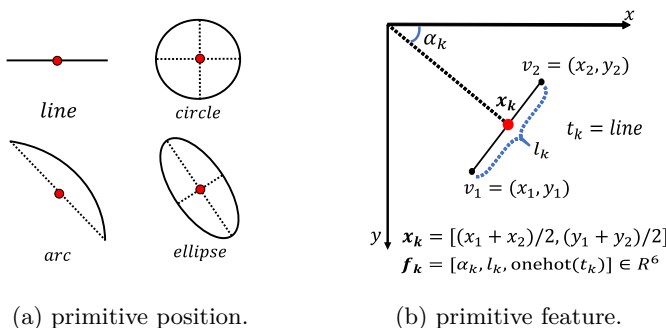

(a) primitive position.    (b) primitive feature.

Figure 1: Illustration of constructing point-based representation.

arranged dense pixels, point clouds are constituted of unordered and irregular point sets. This makes the direct application of image processing methods to point cloud segmentation an impracticable approach. However, in recent years, the integration of neural networks has significantly enhanced the effectiveness of point cloud segmentation across a range of applications, including semantic segmentation (Qi et al., 2017a;a; Zhao et al., 2021), instance segmentation (Ngo et al., 2023; Schult et al., 2023) and panoptic segmentation (Zhou et al., 2021; Li et al., 2022; Hong et al., 2021; Xiao et al., 2023), etc.

## 3 Method

Our methods forgo the raster image or GCN in favor of a point-based representation for graphical primitives. Compared to image-based representations, it reduces the complexity of models due to the sparsity of primitive CAD drawings. In this section, we first describe how to form the point-based representation using the graphical primitives of CAD drawings. Then we illustrate a baseline framework for panoptic symbol spotting. Finally, we thoroughly explain three key techniques, attention with local connection, contrastive connection learning, and KNN interpolation, to adapt this baseline framework to better handle CAD data.

### 3.1 From Symbol to Points

Given vector graphics represented by a set of graphical primitives $\{\boldsymbol{p}_k\}$, we treat it as a collection of points $\{\boldsymbol{p}_k \mid (\boldsymbol{x}_k, \boldsymbol{f}_k)\}$, and each point contains both primitive position $\{\boldsymbol{x}_k\}$ and primitive feature $\{\boldsymbol{f}_k\}$ information; hence, the points set could be unordered and disorganized.

**Primitive position.** Given a graphical primitive, the coordinates of the starting point and the ending point are $(x_1, y_1)$ and $(x_1, y_2)$, respectively. The primitive position $\boldsymbol{x}_k \in \mathbb{R}^2$ is defined as :

$$\boldsymbol{x}_k = [(x_1 + x_2)/2, (y_1 + y_2)/2], \tag{1}$$

We take its center as the primitive position for a closed graphical primitive(circle, ellipse). as shown in Fig. 1a.

**Primitive feature.** We define the primitive features $f_k \in \mathbb{R}^6$ as:

$$\boldsymbol{f}_k = [\alpha_k, l_k, onehot(t_k)], \tag{2}$$

where $\alpha_k$ is the clockwise angle from the $x$ positive axis to $x_k$, and $l_k$ represents the distance between $v_1$ and $v_2$ for linear primitives, as shown in Fig. 1b. For circular primitives like circles and ellipses, $l_k$ is defined as the circumference. We encode the primitive type $t_k$(line, arc, circle, or ellipse) into a one-hot vector to make up the missing information of segment approximations.

### 3.2 Panoptic Symbol Spotting via Point-based Representation

The baseline framework primarily comprises two components: the backbone and the symbol spotting head. The backbone converts raw points into points features, while the symbol

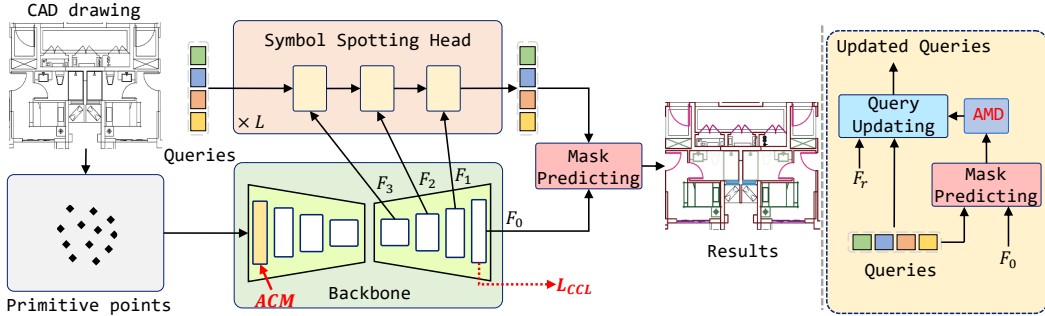

Figure 2: The overview of our method. After transfering CAD drawings to primitive points, we use a backbone to extract multi-resolution features $F_r$ and append a symbol spotting head to spot and recognize symbols. During this process, we propose attention with connection module(ACM), which utilizes primitive connection information when performing self-attention in the first stage of backbone. Subsequently, we propose contrastive connection learning(CCL) to enhance the discriminability between connected primitive features. Finally, we propose KNN interpolation for attention mask downsampling(AMD) to effetively downsample the high-resolution attention masks.

spotting head predicts the symbol mask through learnable queries (Cheng et al., 2021; 2022). Fig. 2 illustrates the the whole framework.

**Backbone.** We choose Point Transformer (Zhao et al., 2021) with a symmetrical encoder and decoder as our backbone for feature extraction due to its good generalization capability in panoptic symbol spotting. The backbone takes primitive points as input, and performs vector attention between each point and its adjacent points to explore local relationships. Given a point $p_i$ and its adjacent points $\mathcal{M}(p_i)$, we project them into query feature $q_i$, key feature $k_j$ and value feature $v_j$, and obtain the vector attention as follows:

$$w_{ij} = \omega(\gamma(q_i, k_j)), \qquad f_i^{\text{attn}} = \sum_{p_j \in \mathcal{M}(p_i)} \text{Softmax}(W_i)_j \odot v_j, \qquad (3)$$

where $\gamma$ serves as a relational function, such as subtraction. $\omega$ is a learnable weight encoding that calculates the attention vectors. $\odot$ is Hadamard product.

**Symbol Spotting Head.** We follow Mask2Former (Cheng et al., 2022) to use hierarchical multi-resolution primitive features $F_r \in \mathbb{R}^{N_r \times D}$ from the decoder of backbone as the input to the symbol spotting predition head, where $N_r$ is the number of feature tokens in resolution $r$ and $D$ is the feature dimension. This head consists of $L$ layers of masked attention modules which progressively upscales low-resolution features from the backbone to produce high-resolution per-pixel embeddings for mask prediction. There are two key components in the masked attention module: *query updating* and *mask predicting*. For each layer $l$, *query updating* involves interacting with different resolution primitive features $F_r$ to update query features. This process can be formulated as,

$$X_l = \text{softmax}(A_{l-1} + Q_l K_l^T)V_l + X_{l-1}, \qquad (4)$$

where $X_l \in \mathbb{R}^{O \times D}$ is the query features. $O$ is the number of query features. $Q_l = f_Q(X_{l-1})$, $K_l = f_K(F_r)$ and $V_l = f_V(F_r)$ are query, key and value features projected by MLP layers. $A_{l-1}$ is the attention mask, which is computed by,

$$A_{l-1}(v) = \begin{cases} 0 & \text{if } M_{l-1}(v) > 0.5, \\ -\infty & \text{otherwise.} \end{cases} \qquad (5)$$

where $v$ is the position of feature point and $M_{l-1}$ is the mask predicted from *mask predicting* part. Note that we need to downsample the high-resolution attention mask to adopt the query updating on low-resolution features. In practice, we utilize four coarse-level primitive features from the decoder of backbone and perform *query updating* from coarse to fine.

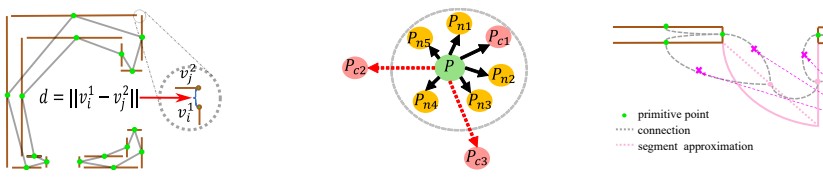

(a) Construct connections.     (b) Attend to connections.     (c) Noisy connections.

Figure 3: (a) Set of primitives and its connection, primitives are disintegrated for clarity. (b) Locally connected primitives are considered in the attention layers. (c) Locally connected primitives do not always belong to the same category.

During *mask predicting* process, we obtain the object mask $M_l \in \mathbb{R}^{O \times N_0}$ and its corresponding category $Y_l \in \mathbb{R}^{O \times C}$ by projecting the query features using two MLP layers $f_Y$ and $f_M$, where $C$ is the category number and $N_0$ is the number of points. The process is as follows:

$$Y_l = f_Y(X_l), \quad M_l = f_M(X_l)F_0^T, \tag{6}$$

The outputs of final layer, $Y_L$ and $M_L$, are the predicted results.

### 3.3 Attention with Connection Module

The simple and unified framework rewards excellent generalization ability by offering a fresh perspective of CAD drawing, a set of points. It can obtain competitive results compared to previous methods. However, it ignores the widespread presence of primitive connections in CAD drawings. It is precisely because of these connections that scattered, unrelated graphical elements come together to form symbols with special semantics. In order to utilize these connections between each primitive, we propose Attention with Connection Module (ACM), the details are shown below.

It is considered that these two graphical primitives$(p_i, p_j)$ are interconnected if the minimum distance $d_{ij}$ between the endpoints $(\boldsymbol{v}_i, \boldsymbol{v}_j)$ of two graphical primitives $(\boldsymbol{p}_i, \boldsymbol{p}_j)$ is below a certain threshold $\epsilon$, where:

$$d_{ij} = \min_{\boldsymbol{v}_i \in \boldsymbol{p}_i, \boldsymbol{v}_j \in \boldsymbol{p}_j} \|\boldsymbol{v}_i - \boldsymbol{v}_j\| < \epsilon. \tag{7}$$

To keep the complexity low, at most $K$ connections are allowed for every graphical primitive by random dropping. Fig. 3a demonstrates the connection construction around the wall symbol, the gray line is the connection between two primitives. In practice, we set $\epsilon$ to 1.0px.

The attention mechanism in (Zhao et al., 2021) directly performs local attention between each point and its adjacent points to explore the relationship. The original attention mechanism interacts only with neighboring points within a spherical region, as shown in Fig. 3b. Our ACM additionally introduces the interaction with locally connected primitive points during attention (pink points), essentially enlarging the radius of the spherical region. Note that we experimentally found that crudely increasing the radius of the spherical region without considering the local connections of primitive points does not result in performance improvement. This may be explained by that enlarging the receptive field also introduces additional noise at the same time. Specifically, we extend the adjacent points set $\mathcal{M}(\boldsymbol{p}_i)$ in Eq. (3) to $\mathcal{A}(\boldsymbol{p}_i) = \mathcal{M}(\boldsymbol{p}_i) \cup \mathcal{C}(\boldsymbol{p}_i)$, where $\mathcal{C}(\boldsymbol{p}_i) = \{\boldsymbol{p}_j | d_{ij} < \epsilon\}$, yielding,

$$f_i^{\mathrm{attn}} = \sum_{\boldsymbol{p}_j \in \mathcal{A}(\boldsymbol{p}_i)} \mathrm{Softmax}(\boldsymbol{W}_i)_j \odot \boldsymbol{v}_j, \tag{8}$$

In practice, since we cannot directly obtain the connection relationships of the points in the intermediate layers of the backbone, we integrate this module into the first stage of the backbone to replace the original local attention, as shown in Fig. 2.

### 3.4 Contrastive Connection Learning.

Although the information of primitive connection are considered when calculating attention of the encoder transformer, locally connected primitives may not belong to the same instance,

in other words, noisy connections could be introduced while take primitive connections into consideration, as shown in Fig. 3c. Therefore, in order to more effectively utilize connection information with category consistency, we follow the widely used InfoNCE loss (Oord et al., 2018) and its generalization (Frosst et al., 2019; Gutmann & Hyvärinen, 2010) to define the contrastive learning objective on the final output feature of backbone. We encourage learned representations more similar to its connected points from the same category and more distinguished from other connected points from different categories. Additionally, we also take neighbor points $\mathcal{M}(\boldsymbol{p}_i)$ into consideration, yielding,

$$L_{CCL} = -\log \frac{\sum_{\boldsymbol{p}_j \in \mathcal{A}(\boldsymbol{p}_i) \wedge l_j = l_i} \exp(-d(\boldsymbol{f}_i, \boldsymbol{f}_j)/\tau)}{\sum_{\boldsymbol{p}_k \in \mathcal{A}(\boldsymbol{p}_i)} \exp(-d(\boldsymbol{f}_i, \boldsymbol{f}_k)/\tau)} \tag{9}$$

where $\boldsymbol{f}_i$ is the backbone feature of $\boldsymbol{p}_i$, $d(\cdot, \cdot)$ is a distance measurement, $\tau$ is the temperature in contrastive learning. we set the $\tau = 1$ by default.

### 3.5 KNN Interpolation

During the process of *query updating* in symbol spotting head Eq. (4) & Eq. (5), we need to convert high-resolution mask predictions to low-resolution for attention masks computation as shown in Fig. 2 (AMD on the right). Mask2Former (Cheng et al., 2022) employs the bilinear interpolation on the pixel-level mask for downsampling. However, the masks of CAD drawings are primitive-level, making it infeasible to directly apply the bilinear interpolation on them. To this end, we propose the KNN interpolation for downsampling the attention masks by fusing the nearest neighbor points. A straightforward operation is max pooling or average pooling. We instead utilize distance-based interpolation. For simplicity, we omit layer index $l$ in $A$,

$$A^r(\boldsymbol{p}_j) = \frac{\sum_{\boldsymbol{p}_j \in \mathcal{K}(\boldsymbol{p}_i)} A^0(\boldsymbol{p}_j)/d(\boldsymbol{p}_i, \boldsymbol{p}_j)}{\sum_{\boldsymbol{p}_j \in \mathcal{K}(\boldsymbol{p}_i)} 1/d(\boldsymbol{p}_i, \boldsymbol{p}_j)} \tag{10}$$

where, $A^0$ and $A^r$ are the full-resolution attention mask and the $r$-resolution attention mask repectively. $d(\cdot, \cdot)$ is a distance measurement. $\mathcal{K}(\boldsymbol{p}_i)$ is the set of $K$ nearest neighbors, In practice, we set $K = 4^r$ in our experiments.

### 3.6 Training and Inference

Throughout the training phase, we adopt bipartite matching and set prediction loss to assign ground truth to predictions with the smallest matching cost. The full loss function $L$ can be formulated as $L = \lambda_{BCE}L_{BCE} + \lambda_{dice}L_{dice} + \lambda_{cls}L_{cls} + \lambda_{CCL}L_{CCL}$, while $L_{BCE}$ is the binary cross-entropy loss (over the foreground and background of that mask), $L_{dice}$ is the Dice loss (Deng et al., 2018), $L_{cls}$ is the default multi-class cross-entropy loss to supervise the queries classification, $L_{CCL}$ is contrastive connection loss. In our experiments, we empirically set $\lambda_{BCE} : \lambda_{dice} : \lambda_{cls} : \lambda_{CCL} = 5 : 5 : 2 : 8$. For inference, we simply use *argmax* to determine the final panoptic results.

## 4 Experiments

In this section, we present the experimental setting and benchmark results on the public CAD drawing dataset FloorPlanCAD (Fan et al., 2021). Following previous works (Fan et al., 2021; Zheng et al., 2022; Fan et al., 2022), we also compare our method with typical image-based instance detection (Ren et al., 2015; Redmon & Farhadi, 2018; Tian et al., 2019; Zhang et al., 2022). Besides, we also compare with point cloud semantic segmentation methods (Zhao et al., 2021), Extensive ablation studies are conducted to validate the effectiveness of the proposed techniques. In addition, we have also validated the generalizability of our method on other datasets beyond floorplanCAD, with detailed results available in the Appendix A.

### 4.1 Experimental Setting

**Dataset and Metrics.** FloorPlanCAD dataset has 11,602 CAD drawings of various floor plans with segment-grained panoptic annotation and covering 30 things and 5 stuff classes.

| Methods | PanCADNet (Fan et al., 2021) | CADTransformer (Fan et al., 2022) | GAT-CADNet (Zheng et al., 2022) | PointT‡ (Zhao et al., 2021) | **SymPoint (ours)** |
|---|---|---|---|---|---|
| F1 | 80.6 | 82.2 | 85.0 | 83.2 | **86.8** |
| wF1 | 79.8 | 80.1 | 82.3 | 80.7 | **85.5** |

Table 1: **Semantic Symbol Spotting** comparison results with previous works. ‡: backbone with double channels. wF1: length-weighted F1.

| Method | Backbone | AP50 | AP75 | mAP | #Params | Speed |
|---|---|---|---|---|---|---|
| FasterRCNN (Ren et al., 2015) | R101 | 60.2 | 51.0 | 45.2 | 61M | 59ms |
| YOLOv3 (Redmon & Farhadi, 2018) | DarkNet53 | 63.9 | 45.2 | 41.3 | 62M | 11ms |
| FCOS (Tian et al., 2019) | R101 | 62.4 | 49.1 | 45.3 | 51M | 57ms |
| DINO (Zhang et al., 2022) | R50 | 64.0 | 54.9 | 47.5 | 47M | 42ms |
| **SymPoint (ours)** | PointT‡ | **66.3** | **55.7** | **52.8** | 35M | 66ms |

Table 2: **Instance Symbol Spotting** comparison results with image-based detection methods.

Following (Fan et al., 2021; Zheng et al., 2022; Fan et al., 2022), we use the panoptic quality (PQ) defined on vector graphics as our main metric to evaluate the performance of panoptic symbol spotting. By denoting a graphical primitive $e = (l, z)$ with a semantic label $l$ and an instance index $z$, PQ is defined as the multiplication of segmentation quality (SQ) and recognition quality (RQ), which is formulated as

$$PQ = RQ \times SQ \tag{11}$$

$$= \frac{|TP|}{|TP| + \frac{1}{2}|FP| + \frac{1}{2}|FN|} \times \frac{\sum_{(s_p, s_g) \in TP} \text{IoU}(s_p, s_g)}{|TP|} \tag{12}$$

$$= \frac{\sum_{(s_p, s_g) \in TP} \text{IoU}(s_p, s_g)}{|TP| + \frac{1}{2}|FP| + \frac{1}{2}|FN|}. \tag{13}$$

where, $s_p = (l_p, z_p)$ is the predicted symbol, $s_g = (l_g, z_g)$ is the ground truth symbol. $|TP|$, $|FP|$, $|FN|$ indicate true positive, false positive and false negative respectively. A certain predicted symbol is considered as matched if it finds a ground truth symbol, with $l_p = l_g$ and $\text{IoU}(s_p, s_g) > 0.5$, where the IoU is computed by:

$$\text{IoU}(s_p, s_g) = \frac{\Sigma_{e_i \in s_p \cap s_g} log(1 + L(e_i))}{\Sigma_{e_j \in s_p \cup s_g} log(1 + L(e_j))}. \tag{14}$$

**Implementation Details.** We implement SymPoint with Pytorch. We use PointT (Zhao et al., 2021) with double channels as the backbone and stack $L = 3$ layers for the symbol spotting head. For data augmentation, we adopt rotation, flip, scale, shift, and cutmix augmentation. We choose AdamW (Loshchilov & Hutter, 2017) as the optimizer with a default weight decay of 0.001, the initial learning rate is 0.0001, and we train the model for 1000 epochs with a batch size of 2 per GPU on 8 NVIDIA A100 GPUs.

## 4.2 Benchmark Results

**Semantic symbol spotting.** We compare our methods with point cloud segmentation methods (Zhao et al., 2021), and symbol spotting methods (Fan et al., 2021; 2022; Zheng et al., 2022). The main test results are summarized in Tab. 1, Our algorithm surpasses all previous methods in the task of semantic symbol spotting. More importantly, compared to GAT-CADNet (Zheng et al., 2022), we achieves an absolute improvement of **1.8% F1.** and **3.2% wF1** respectively. For the PointT‡, we use our proposed point-based representation in Section 3.1 to convert the CAD drawing to a collection of points as input. It is worth noting that PointT‡ has already achieved comparable results to GAT-CADNet (Zheng et al., 2022), which demonstrates the effectiveness of the proposed point-based representation for CAD symbol spotting.

**Instance Symbol Spotting.** We compare our method with various image detection methods, including FasterRCNN (Ren et al., 2015), YOLOv3 (Redmon & Farhadi, 2018),

| Method | Data Format | PQ | SQ | RQ | #Params | Speed |
|---|---|---|---|---|---|---|
| PanCADNet (Fan et al., 2021) | VG + RG | 55.3 | 83.8 | 66.0 | >42M | >1.2s |
| CADTransformer (Fan et al., 2022) | VG + RG | 68.9 | 88.3 | 73.3 | >65M | >1.2s |
| GAT-CADNet (Zheng et al., 2022) | VG | 73.7 | 91.4 | 80.7 | - | - |
| PointT‡Cluster(Zhao et al., 2021) | VG | 49.8 | 85.6 | 58.2 | 31M | 80ms |
| **SymPoint(ours, 300epoch)** | VG | **79.6** | 89.4 | **89.0** | 35M | 66ms |
| **SymPoint(ours, 500epoch)** | VG | **81.9** | 90.6 | **90.4** | 35M | 66ms |
| **SymPoint(ours, 1000epoch)** | VG | **83.3** | **91.4** | **91.1** | 35M | 66ms |

Table 3: **Panoptic Symbol Spotting** comparisons results with previous works. VG: vector graphics, RG: raster graphics.

| Baseline | ACM | CCL | KInter | PQ | RQ | SQ |
|---|---|---|---|---|---|---|
| ✓ | | | | 73.1 | 83.3 | 87.7 |
| ✓ | ✓ | | | 72.6 | 82.9 | 87.6 |
| ✓ | | ✓ | | 73.5 | 83.9 | 87.6 |
| ✓ | ✓ | ✓ | | 74.3 | 85.8 | 86.6 |
| ✓ | ✓ | ✓ | ✓ | 77.3 | 87.1 | 88.7 |

(a) Ablation studies of different techniques

| DSampling method | PQ | RQ | SQ |
|---|---|---|---|
| linear | 74.3 | 85.8 | 86.6 |
| knn avepool | 75.9 | 85.9 | 88.4 |
| knn maxpool | 77.0 | 86.7 | 88.8 |
| knn interp | 77.3 | 87.1 | 88.7 |

(b) Ablation studies of mask downsampling

| | BS | SW | L | O | D | PQ | RQ | SQ | IoU | #Params |
|---|---|---|---|---|---|---|---|---|---|---|
| | 1x | ✓ | 3 | 300 | 128 | 67.1 | 78.7 | 85.2 | 62.8 | 9M |
| | 1.5x | ✓ | 3 | 300 | 128 | 73.3 | 84.0 | 87.3 | 65.6 | 19M |
| ablation setting | 2x | ✓ | 3 | 300 | 128 | 77.3 | 87.1 | 88.7 | 68.3 | 32M |
| | 2x | ✓ | 3 | 500 | 128 | 77.9 | 87.6 | 88.9 | 68.8 | 32M |
| final setting | 2x | ✓ | 3 | 500 | 256 | 79.6 | 89.0 | 89.4 | 69.1 | 35M |
| | 2x | | 3 | 500 | 256 | 79.1 | 88.4 | 89.5 | 68.8 | 42M |
| | 2x | ✓ | 6 | 500 | 256 | 79.0 | 88.1 | 89.6 | 68.5 | 35M |

(c) Ablation studies on architecture design. BS: Backbone size. SW: share weights. *L*: layer number of spotting head. *O*: query number. *D*: feature dimension. ✓ in the share weights column means whether share weights for head layers.

Table 4: **Ablation Stuides** on different techniques, attention mask downsampling, and architecture desgin.

FCOS (Tian et al., 2019), and recent DINO (Zhang et al., 2022). For a fair comparison, we post-process the predicted mask to produce a bounding box for metric computation. The main comparison results are listed in Tab. 2. Although our framework is not trained to output a bounding box, it still achieves the best results.

**Panoptic Symbol Spotting.** To verify the effectiveness of the symbol spotting head, we also design a variant method without this head, named PointT‡Cluster, which predicts an offset vector per graphic entity to gather the instance entities around a common instance centroid and performs class-wise clustering (e.g. meanshift (Cheng, 1995)) to get instance labels as in CADTransformer (Fan et al., 2022). The final results are listed in Tab. 3. Our SymPoint trained with 300epoch outperforms both PointT‡Cluster and the recent SOTA method GAT-CADNet(Zheng et al., 2022) substantially, demonstrate the effectiveness of the proposed method. Our method also benefits from longer training and achieves further performance improvement. What's more, our method runs much faster during the inference phase than previous methods. For image-based methods, it takes approximately 1.2s to render a vector graphic into an image while our method does not need this process. The qualitative results are shown in Fig. 4.

## 4.3 ABLATION STUDIES

In this section, we carry out a series of comprehensive ablation studies to clearly illustrate the potency and intricate details of the SymPoint framework. All ablations are conducted under 300 epoch training.

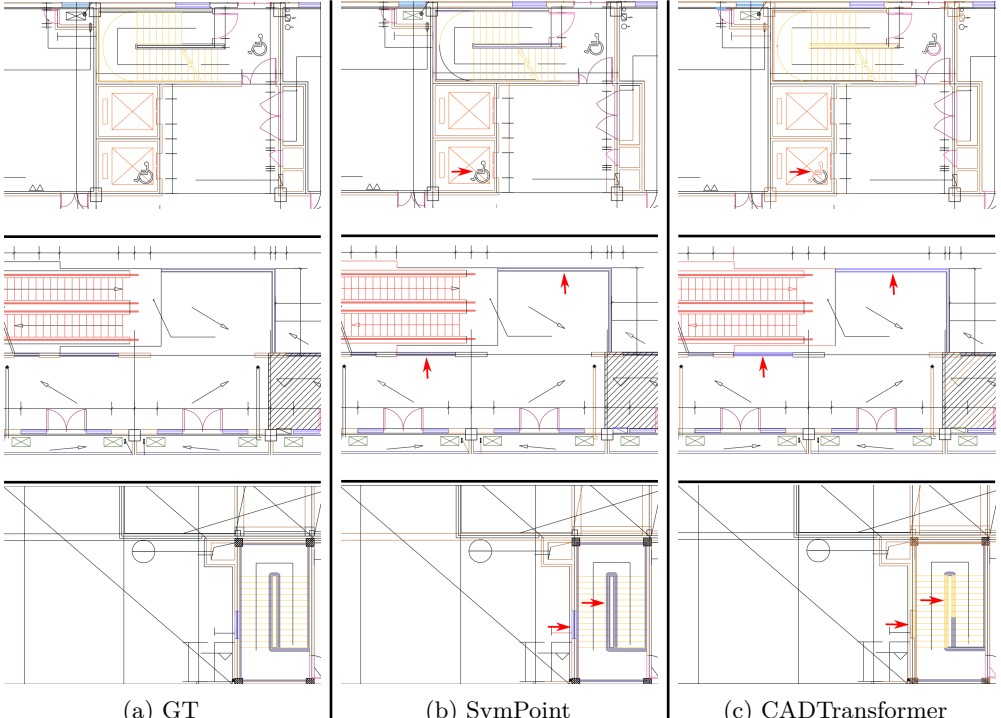

|      (a) GT      |   (b) SymPoint   | (c) CADTransformer |

Figure 4: Qualitative comparison of panoptic symbol spotting results with CADTransformer. Primitives belonging to different classes are represented in distinct colors. The colormap for each category can be referenced in Fig. 8.

**Effects of Techniques.** We conduct various controlled experiments to verify different techniques that improve the performance of SymPoint in Tab. 4a. Here the baseline means the method described in Sec. 3.2. When we only introduce ACM (Attention with Connection Module), the performance drops a bit due to the noisy connections. But when we combine it with CCL (Contrastive Connection Learning), the performance improves to 74.3 of PQ. Note that applying CCL alone could only improve the performance marginally. Furthermore, KNN Interpolation boosts the performance significantly, reaching 77.3 of PQ.

**KNN Interpolation.** In Tab. 4b, we ablate different ways of downsampling attention mask: 1) linear interpolation, 2) KNN average pooling, 3) KNN max pooling, 4) KNN interpolation. KNN average pooling and KNN max pooling means using the averaged value or max value of the K nearest neighboring points as output instead of the one defined in Eq. (10). We can see that the proposed KNN interpolation achieves the best performance.

**Architecture Design.** We analyze the effect of varying model architecture design, like channel number of backbone and whether share weights for the L layers of symbol spotting head. As shown in Tab. 4c, we can see that enlarging the backbone, the query number and the feature channels of the symbol spotting head could further improve the performance. Sharing weights for spotting head not only saves model parameters but also achieves better performance compared with the one that does not share weights.

## 5 CONCLUSION AND FUTURE WORK

This work introduces a novel perspective for panoptic symbol spotting. We treat CAD drawings as sets of points and utilize methodologies from point cloud analysis for symbol spotting. Our method SymPoint is simple yet effective and outperforms previous works. One limitation is that our method needs a long training epoch to get promising performance. Thus accelerating model convergence is an important direction for future work.

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

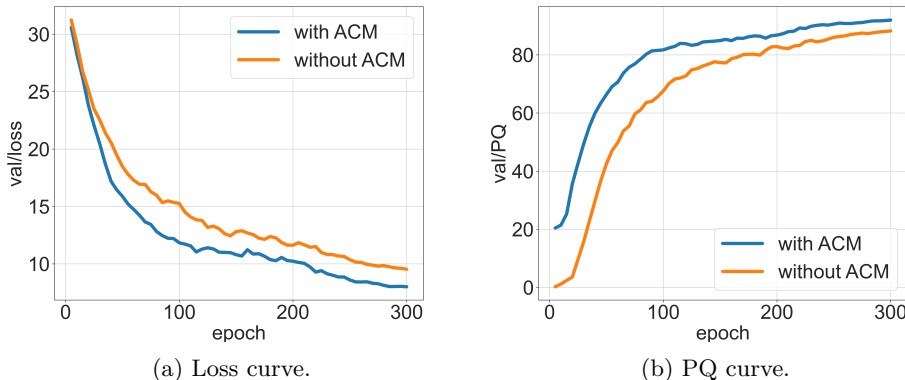

(a) Loss curve.   (b) PQ curve.

Figure 5: Convergence curves with/without the ACM Module on SESYD-floorplans.

# A  APPENDIX

Due to space constraints in the paper,additional techniques analysis, additional quantitative results, qualitative results, and other dataset results can be found in the supplementary materials.

## A.1  ADDITIONAL TECHNIQUES ANALYSIS

**Effects of Attention with Connection Module.**   We conduct additional experiments in SESYD-floorplans dataset that is smaller than floorplanCAD. ACM can significantly promote performance and accelerate model convergence. We present the convergence curves without/with ACM in Fig. 5.

**Explanation and Visualization of KNN interpolation Technique.**   While bilinear interpolation, as utilized in Mask2Former, is tailored for regular data, such as image, but it is unsuitable for irregular sparse primitive points. Here, we provided some visualizations of point masks for KNN interpolation and bilinear interpolation as shown in Fig. 6. Note that these point masks are soft masks (ranging from 0 to 1) predicted by intermediate layers. After downsampling the point mask to 4x and 16x, we can clearly find that KNN interpolation well preserves the original mask information, while bilinear interpolation causes a significant information loss, which could harm the final performance.

## A.2  ADDITIONAL QUANTITATIVE EVALUATIONS

We present a detailed evaluation of panoptic quality(PQ), segmentation quality(SQ), and recognition quality(RQ) in Tab. 5. Here, we provide the class-wise evaluations of different variants of our methods.

## A.3  ADDITIONAL DATASETS

To demonstrate the generality of our SymPoint, we conducted experiments on other datasets beyond floorplanCAD.

**Private Dataset.**   We have also collected a dataset of floorplan CAD drawings with 14,700 from our partners. We've meticulously annotated the dataset at the primitive level. Due to privacy concerns, this dataset is currently not publicly available. we randomly selected 10,200 as the training set and the remaining 4,500 as the test set. We conduct ablation studies of the proposed three techniques on this dataset, and the results are shown in Tab. 6. Different from the main paper, we also utilize the color information during constructing the connections, *i.e.*, locally connected primitives with the same color are considered as

| Class | A | | | B | | | C | | | D | | | E | | |
|---|---|---|---|---|---|---|---|---|---|---|---|---|---|---|---|
| | PQ | RQ | SQ | PQ | RQ | SQ | PQ | RQ | SQ | PQ | RQ | SQ | PQ | RQ | SQ |
| single door | 83.2 | 90.8 | 91.6 | 82.9 | 90.7 | 91.4 | 83.3 | 90.9 | 91.6 | 86.6 | 93.0 | 93.1 | 91.7 | 96.0 | 95.5 |
| double door | 86.8 | 93.8 | 92.5 | 85.8 | 93.4 | 91.9 | 86.5 | 93.5 | 92.5 | 88.5 | 95.3 | 92.9 | 91.5 | 96.6 | 94.7 |
| sliding door | 87.5 | 94.0 | 93.1 | 87.6 | 94.6 | 92.5 | 88.6 | 94.4 | 93.8 | 90.4 | 96.4 | 93.7 | 94.8 | 97.7 | 97.0 |
| folding door | 39.3 | 46.8 | 84.0 | 48.5 | 58.2 | 83.3 | 44.7 | 55.2 | 80.9 | 56.4 | 61.5 | 91.6 | 73.8 | 87.0 | 84.8 |
| revolving door | 0.0 | 0.0 | 0.0 | 0.0 | 0.0 | 0.0 | 0.0 | 0.0 | 0.0 | 0.0 | 0.0 | 0.0 | 0.0 | 0.0 | 0.0 |
| rolling door | 0.0 | 0.0 | 0.0 | 0.0 | 0.0 | 0.0 | 0.0 | 0.0 | 0.0 | 0.0 | 0.0 | 0.0 | 0.0 | 0.0 | 0.0 |
| window | 64.5 | 77.1 | 83.6 | 67.2 | 80.7 | 83.2 | 69.6 | 83.0 | 83.8 | 73.0 | 85.8 | 85.0 | 78.9 | 90.4 | 87.3 |
| bay window | 6.8 | 10.7 | 63.8 | 3.4 | 5.3 | 63.4 | 4.6 | 7.4 | 62.0 | 11.4 | 16.9 | 67.8 | 35.4 | 42.3 | 83.6 |
| blind window | 73.3 | 86.9 | 84.4 | 69.8 | 87.2 | 80.1 | 70.6 | 86.3 | 81.8 | 74.0 | 89.7 | 82.5 | 80.6 | 92.1 | 87.5 |
| opening symbol | 16.3 | 24.5 | 66.5 | 30.1 | 40.1 | 75.1 | 11.3 | 17.0 | 66.7 | 20.7 | 29.9 | 69.2 | 33.1 | 40.9 | 80.7 |
| sofa | 69.4 | 78.1 | 88.9 | 67.4 | 79.0 | 85.3 | 70.4 | 79.4 | 88.7 | 74.0 | 83.5 | 88.6 | 83.9 | 88.8 | 94.5 |
| bed | 74.7 | 89.3 | 83.6 | 70.5 | 85.9 | 82.1 | 73.4 | 87.9 | 83.5 | 73.2 | 87.4 | 83.7 | 86.1 | 95.9 | 89.8 |
| chair | 69.6 | 76.4 | 91.1 | 74.0 | 81.5 | 90.7 | 68.5 | 76.5 | 89.6 | 75.9 | 82.7 | 91.7 | 82.7 | 88.9 | 93.1 |
| table | 55.4 | 66.1 | 83.7 | 53.3 | 64.4 | 82.8 | 49.3 | 59.9 | 82.4 | 60.2 | 70.2 | 85.7 | 70.9 | 79.1 | 89.6 |
| TV cabinet | 72.7 | 87.1 | 83.5 | 65.5 | 82.8 | 79.2 | 72.6 | 88.1 | 82.4 | 80.1 | 92.9 | 86.3 | 90.1 | 97.0 | 92.9 |
| Wardrobe | 78.3 | 93.3 | 83.9 | 79.8 | 94.8 | 84.1 | 80.3 | 94.7 | 84.8 | 83.0 | 95.4 | 87.0 | 87.7 | 96.4 | 90.9 |
| cabinet | 59.1 | 69.7 | 84.8 | 63.6 | 75.9 | 83.7 | 63.2 | 75.3 | 83.9 | 67.5 | 80.5 | 83.9 | 73.8 | 86.2 | 85.6 |
| gas stove | 96.4 | 98.8 | 97.6 | 95.2 | 98.9 | 96.3 | 97.1 | 98.9 | 98.2 | 97.5 | 99.3 | 98.2 | 97.6 | 98.9 | 98.7 |
| sink | 80.1 | 89.7 | 89.3 | 81.5 | 91.2 | 89.3 | 81.9 | 91.0 | 89.9 | 83.3 | 91.8 | 90.8 | 86.1 | 92.9 | 92.7 |
| refrigerator | 75.9 | 91.2 | 83.2 | 73.9 | 90.7 | 81.5 | 75.1 | 91.4 | 82.2 | 79.4 | 94.0 | 84.5 | 87.8 | 95.7 | 91.8 |
| airconditioner | 68.6 | 75.9 | 90.4 | 67.8 | 77.7 | 87.3 | 66.7 | 75.3 | 88.6 | 73.7 | 80.1 | 92.0 | 80.5 | 84.4 | 95.4 |
| bath | 57.2 | 75.0 | 76.3 | 54.0 | 71.1 | 75.9 | 60.7 | 77.7 | 78.1 | 64.6 | 81.7 | 79.1 | 73.2 | 85.0 | 86.1 |
| bath tub | 62.7 | 78.8 | 79.6 | 57.4 | 78.3 | 73.2 | 61.6 | 80.3 | 76.7 | 65.0 | 82.3 | 79.0 | 76.1 | 91.4 | 83.2 |
| washing machine | 74.0 | 86.3 | 85.8 | 73.4 | 88.8 | 82.6 | 78.1 | 90.3 | 86.5 | 78.2 | 90.7 | 86.2 | 86.7 | 93.8 | 92.5 |
| urinal | 89.2 | 93.5 | 95.5 | 90.0 | 95.0 | 94.7 | 89.5 | 94.3 | 94.9 | 91.4 | 96.0 | 95.2 | 93.8 | 96.7 | 96.9 |
| squat toilet | 88.8 | 95.7 | 92.7 | 90.2 | 96.6 | 93.4 | 89.6 | 96.0 | 93.4 | 90.4 | 96.8 | 93.3 | 93.6 | 97.5 | 96.1 |
| toilet | 86.5 | 94.4 | 91.6 | 88.4 | 96.0 | 92.1 | 88.8 | 96.4 | 92.1 | 90.0 | 96.0 | 93.7 | 92.9 | 97.2 | 95.6 |
| stairs | 61.3 | 77.8 | 78.8 | 60.9 | 77.9 | 78.1 | 61.4 | 79.1 | 77.6 | 64.9 | 82.1 | 79.1 | 72.5 | 85.3 | 85.0 |
| elevator | 79.8 | 89.6 | 89.0 | 75.8 | 89.1 | 85.2 | 76.4 | 87.9 | 86.9 | 81.5 | 91.3 | 89.2 | 88.8 | 94.4 | 94.1 |
| escalator | 33.9 | 48.1 | 70.6 | 35.4 | 49.6 | 71.4 | 33.6 | 50.0 | 67.2 | 51.4 | 68.5 | 75.0 | 60.6 | 75.6 | 80.2 |
| row chairs | 85.1 | 90.8 | 93.8 | 80.2 | 86.6 | 92.7 | 84.6 | 90.6 | 93.3 | 83.9 | 89.9 | 93.4 | 84.3 | 89.2 | 94.5 |
| parking spot | 65.8 | 80.2 | 82.1 | 65.3 | 81.4 | 80.2 | 68.9 | 85.7 | 80.4 | 70.4 | 84.5 | 83.3 | 73.4 | 86.7 | 84.7 |
| wall | 35.7 | 55.5 | 64.4 | 31.7 | 51.1 | 62.2 | 41.6 | 64.6 | 64.4 | 38.5 | 59.4 | 64.9 | 53.5 | 77.5 | 69.0 |
| curtain wall | 30.1 | 41.6 | 72.3 | 34.4 | 46.2 | 74.4 | 31.5 | 44.2 | 71.2 | 33.8 | 46.1 | 73.3 | 44.2 | 60.2 | 73.5 |
| railing | 20.7 | 29.0 | 71.4 | 16.1 | 23.2 | 69.5 | 28.1 | 37.7 | 74.5 | 29.7 | 40.9 | 72.5 | 53.0 | 66.3 | 80.0 |
| total | 73.1 | 83.3 | 87.7 | 72.7 | 82.9 | 87.6 | 74.3 | 85.8 | 86.6 | 77.3 | 87.1 | 88.7 | 83.3 | 91.1 | 91.4 |

Table 5: **Quantitative results for panoptic symbol spotting** of each class. In the test split, some classes have a limited number of instances, resulting in zeros and notably low values in the results. **A**: Baseline. **B**: Baseline+ACM. **C**: Baseline+ACM+CCL. **D**: Baseline+ACM+CCL+KInter. **E**: Final setting + long training epoch.

| Baseline | ACM | CCL | KInter | PQ | RQ | SQ |
|---|---|---|---|---|---|---|
| ✓ | | | ✓ | 62.1 | 75.3 | 82.4 |
| ✓ | ✓ | | ✓ | 64.9 | 76.1 | 85.3 |
| ✓ | ✓ | ✓ | ✓ | 66.7 | 77.3 | 86.4 |
| ✓ | ✓ | ✓ | | 62.5 | 74.3 | 84.1 |

Table 6: **Ablation Stuides** on different techniques in private dataset.

valid connections. We do not use color information in the floorCAD dataset because their color information is not consistent for the same category while ours is consistent. It can be seen that applying ACM does not lead to a decline in performance. In fact, there's an approximate 3% improvement in the PQ.

**Vector Graphics Recognition Dataset.** Similar to (Jiang et al., 2021; Shi et al., 2022), we evaluate our method on SESYD, a public dataset comprising various types of vector graphic documents. This database is equipped with object detection ground truth. For our experiments, we specifically focused on the **floorplans** and **diagrams** collections. The results are presented in Tab. 7 We achieved results on par with YOLaT(Jiang et al., 2021) and RendNet(Shi et al., 2022), which are specifically tailored for detection tasks. The aforementioned results further underscore the robust generalizability of our method. Some comparison visualized results with YOLaT are shown in Fig. 7.

| Methods | AP50 | AP75 | mAP |
|---------|------|------|-----|
| Yolov4 | 93.04 | 87.48 | 79.59 |
| YOLaT | 98.83 | 94.65 | 90.59 |
| RendNet | 98.70 | 98.25 | 91.37 |
| SymPoint | 96.79 | 95.63 | 91.01 |

(a) Performance comparison on floorplans.

| Methods | AP50 | AP75 | mAP |
|---------|------|------|-----|
| Yolov4 | 88.71 | 84.65 | 76.28 |
| YOLaT | 96.63 | 94.89 | 89.67 |
| RendNet | - | - | - |
| SymPoint | 97.0 | 94.51 | 90.26 |

(b) Performance comparison on diagrams.

Table 7: **Performance comparison** on floorplans and diagrams.

## A.4 ADDITIONAL QUALITATIVE EVALUATIONS

The results of additional cases are visually represented in this section, you can zoom in on each picture to capture more details, primitives belonging to different classes are represented in distinct colors. The color representations for each category can be referenced in Fig. 8. Some visualized results are shown in Fig. 9, Fig. 10 and Fig. 11 .

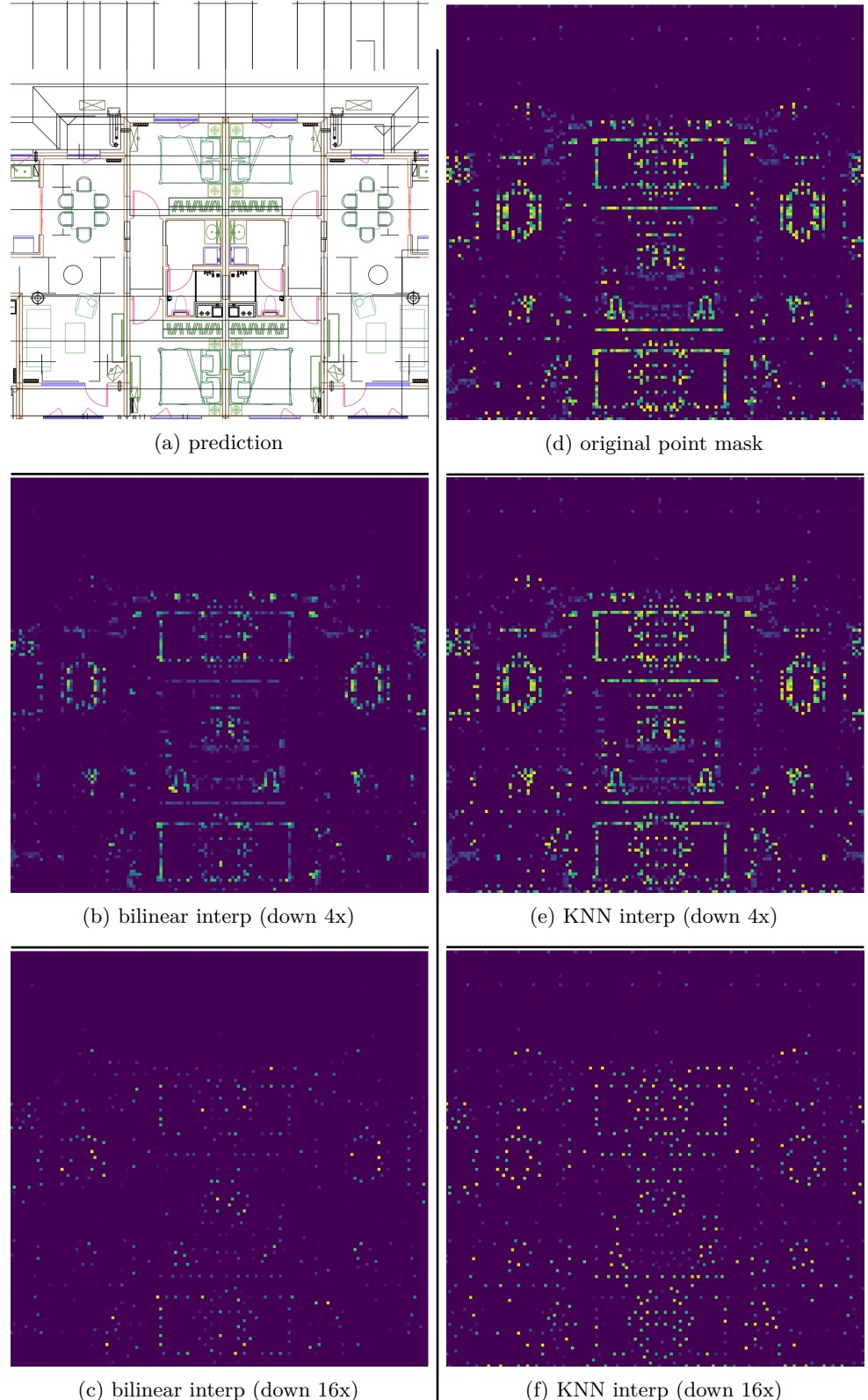

(a) prediction

(d) original point mask

(b) bilinear interp (down 4x)

(e) KNN interp (down 4x)

(c) bilinear interp (down 16x)

(f) KNN interp (down 16x)

Figure 6: KNN interp vs bilinear interp.

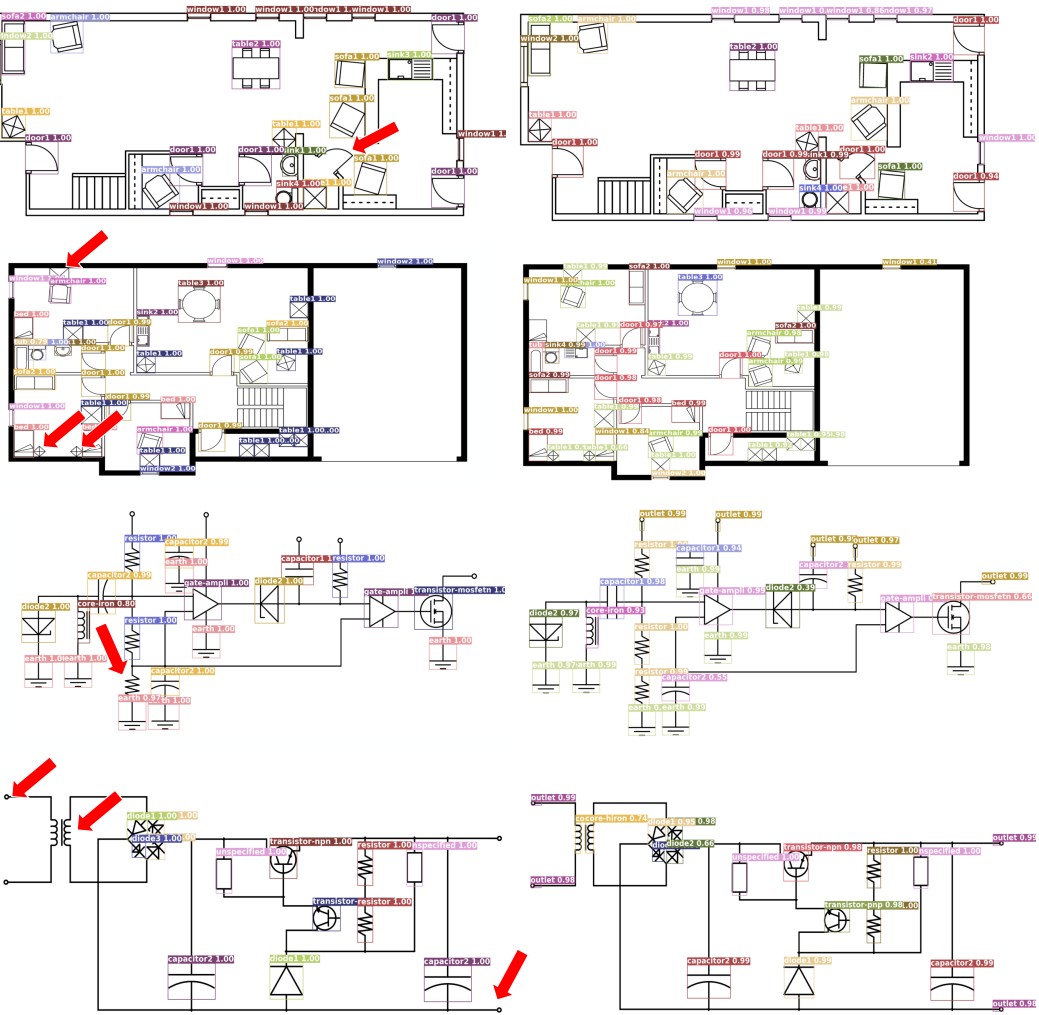

Figure 7: Qualitative comparison on floorplans and diagrams with YOLaT. The left column displays YOLaT's results, while the right column showcases ours.

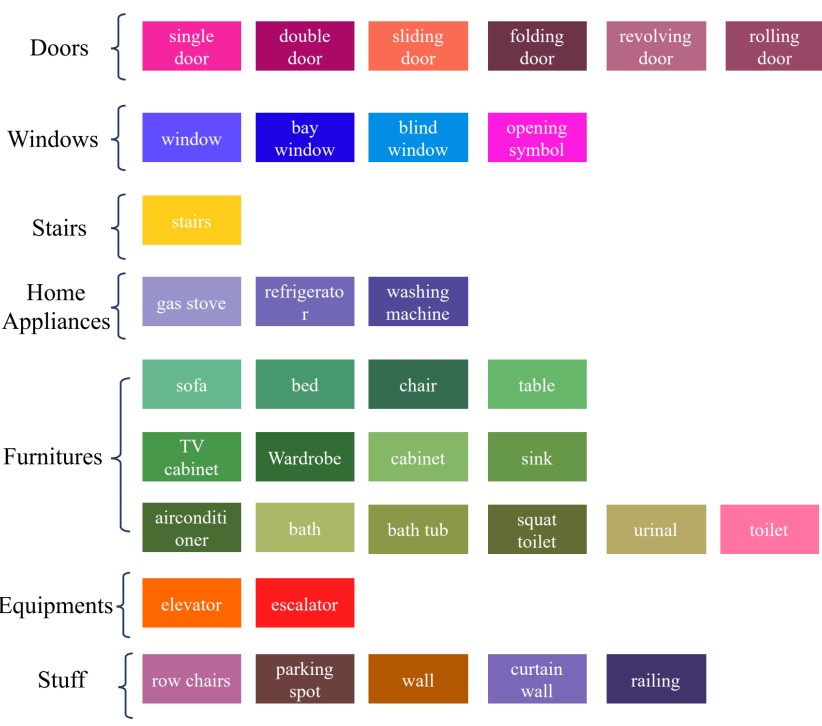

Figure 8: A visualized color map is provided for each class along with its corresponding super-class.

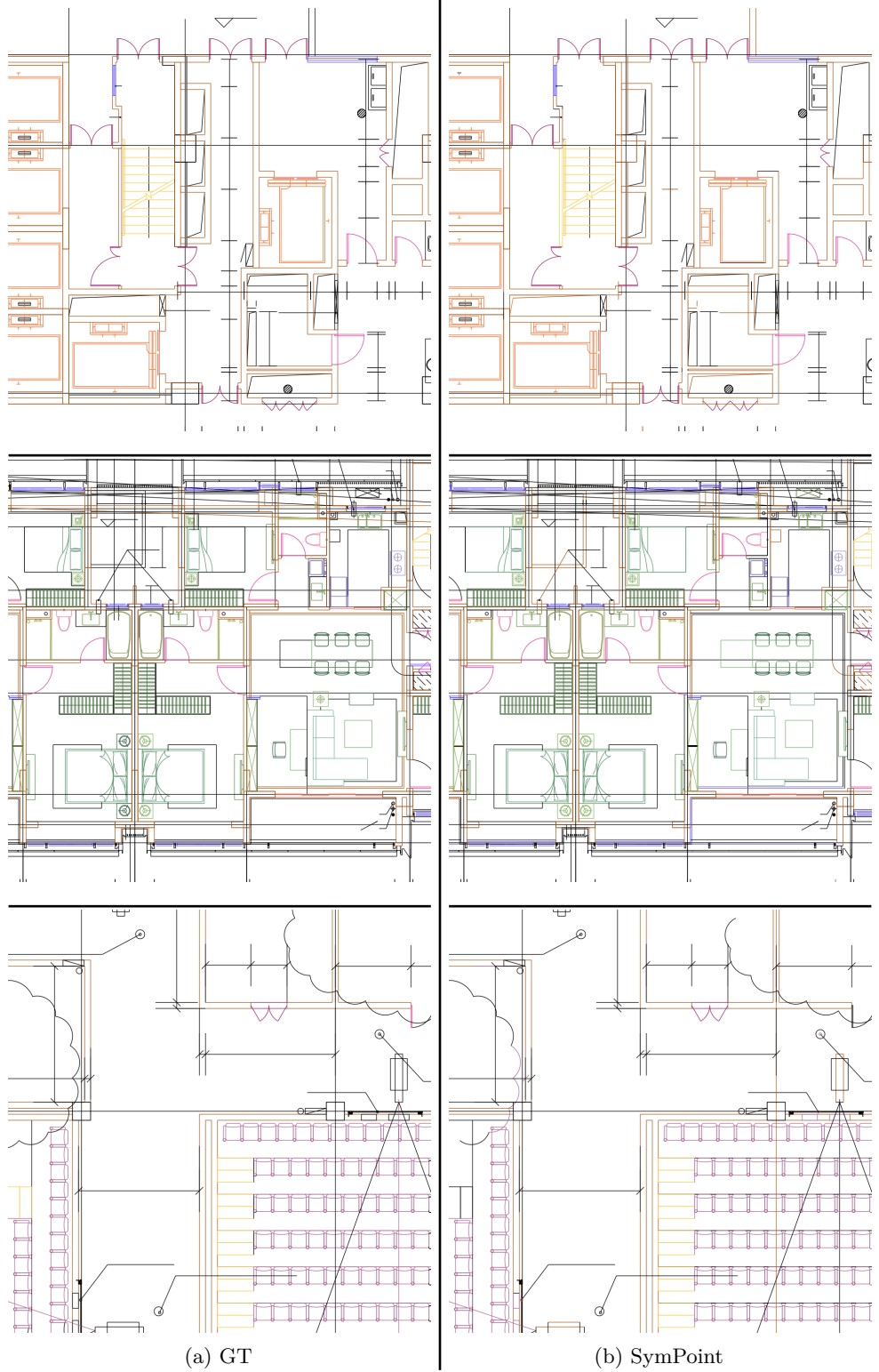

(a) GT                                     (b) SymPoint

Figure 9: Results of SymPoint on FloorPlanCAD.

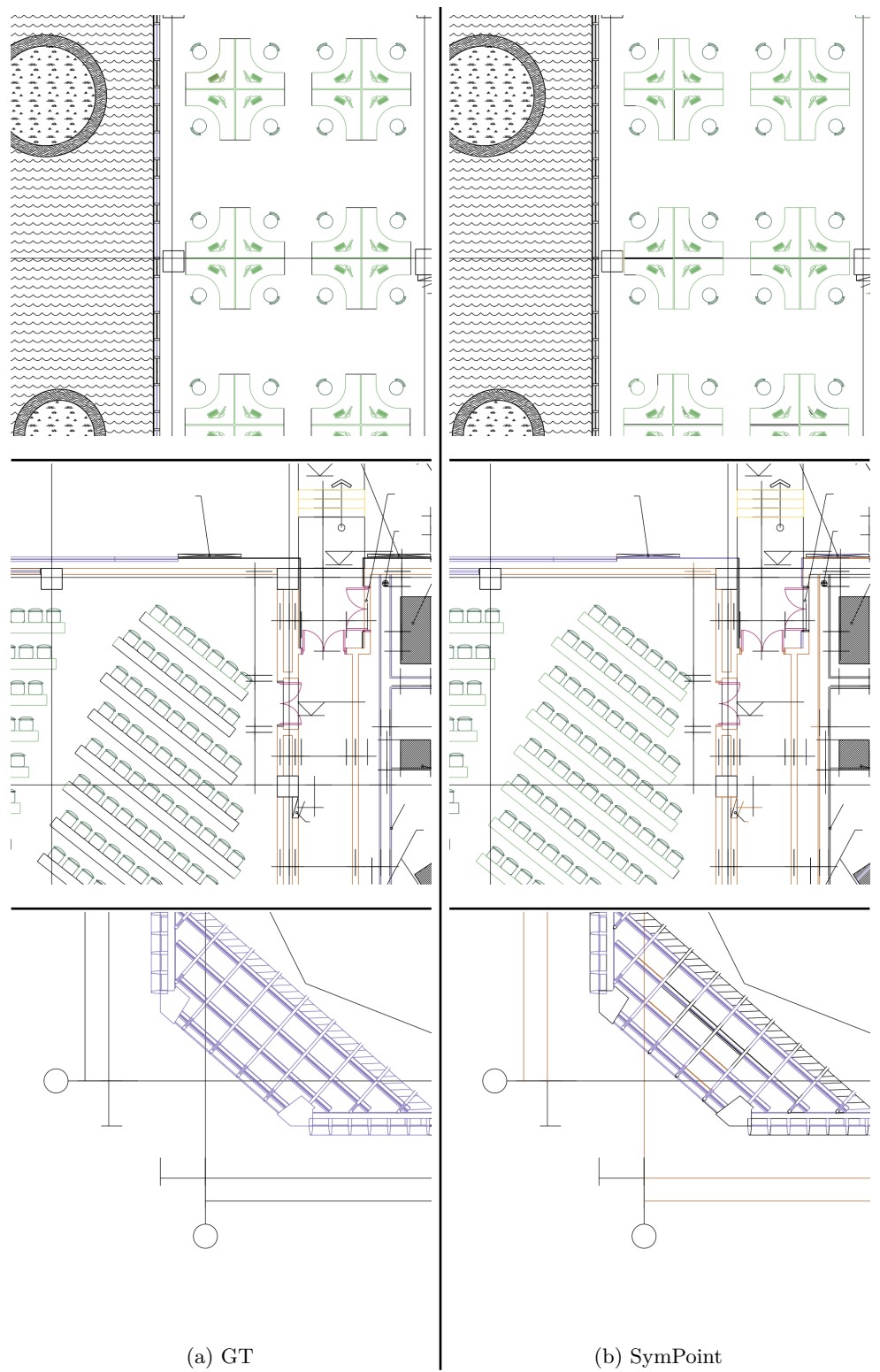

(a) GT

(b) SymPoint

Figure 10: Results of SymPoint on FloorPlanCAD.

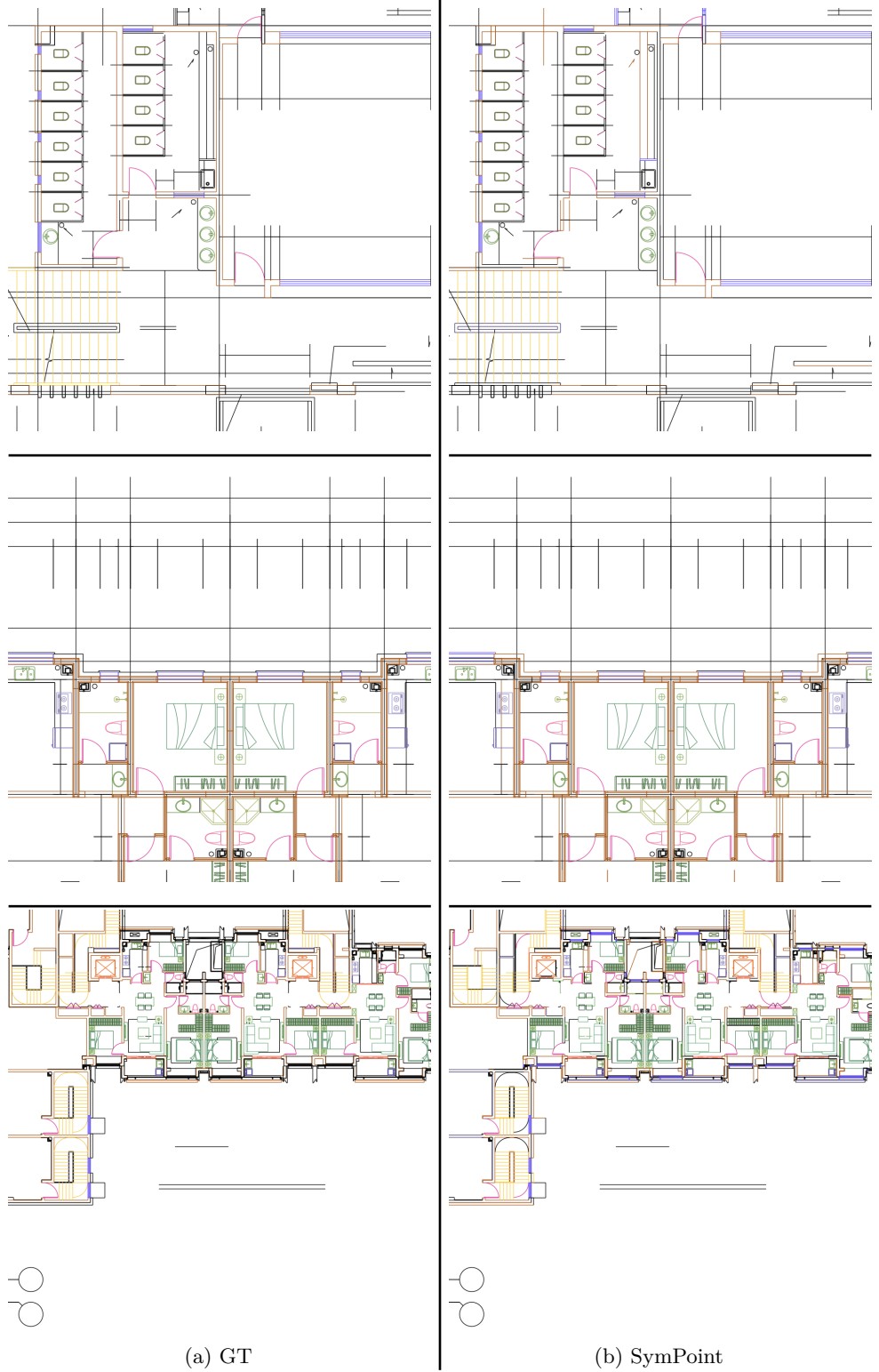

(a) GT          (b) SymPoint

Figure 11: Results of SymPoint on FloorPlanCAD.

