# OpenReview forum: "Symbol as Points: Panoptic Symbol Spotting via Point-based Representation"
_ICLR.cc/2024/Conference — ICLR 2024 poster_

### Official Review · Reviewer_q1T8 · 2023-10-31

**Soundness:** 3 good
**Presentation:** 3 good
**Contribution:** 2 fair
**Rating:** 6
**Confidence:** 3

**Summary:**

The paper proposes a new method for symbol segmentation in architectural floorplans. The method is based on representing each graphical primitive as a point with a set of features and thus, relying on Point Transformer for feature extraction. Then, an adaptation of Mask2Former is used to segment and classify the symbols in the floorplan. Some specific components are introduced to adress the specificity of graphical primitives in architectural drawings. Experimental validation is performed by applying the method to a standard floorplan dataset, comparing with state-of-the-art and conducting several ablation studies.

**Strengths:**

- The idea of relying on a point representation of graphical primitives seems novel and makes sense in this context since symbols in architectural drawings are composed of graphical primitives. Then, using the combination of PointTransformer and Mask2Former is also a novel approach in this context that seems suitable for capturing the interaction between graphical primitives for symbol segmentation.
- Experimental results on a standard dataset report better performance than state-of-the-art methods. A detailed ablation study shows the contribution of the different modules of the proposed framework.

**Weaknesses:**

Perhaps I missunderstood something, but I do not see the motivation of symbol segmentation in CAD drawings. As far as I understand CAD drawings should already contain information about the symbols included in the floorplan and where they are located.

In the description of the method and the experiments there are several points that are confusing or not well explained:
- In equation (2) I understand that l_k is the distance between v_1 and v_2. Then, what about circles and ellipses? How is the lengh computed? And for arcs, this definition does not account for the curvature. Two arcs with very different curvature can have the same representation.
- In equation (3), it is not clear how the neighbourhood M(p_i) is defined. Do adjacent points mean connected primitives? Or primitives inside a certain distance? Which is exactly the difference with A(p_i) defined later in section 3.3 (given that the threshold used in section 3.3 is just one pixel). In this sense, the role of the ACM module is not very clear.
- It is not clear the motivation of the KNN interpolation described in section 3.5. As far as I understand, since points correspond to graphical primitives, interpolation of neighboring points could lead to losing information of specific primitives and I am not sure that makes sense merging different primitives into a new one.
- Related to the previous point, It is not clear how it is performed downsamplind and upsampling in the Point Transformer. The same as in the original Point Transformer?
- In equation (12) it is not clear what is e_i and L(e_i).
- In the experiments, which is the difference between Semantic and Panoptic Symbol Segmentation? Why in table 1 (semantic segmentation) the evaluation measure is F1? How are F1 and wF1 defined in this context?

**Questions:**

See above in Weaknesses

---

> ### Author Response · Authors · 2023-11-16
>
> Dear Reviewer q1T8,
>
> Thank you very much for your comment and support. We address your concerns as follows.
>
> ### 1. Motivation for Symbol Segmentation
>
> While tools like AutoCAD enable the rapid creation of  CAD drawing, it typically lack detailed categorization information for each graphical primitive or symbol. The task of  panoptic symbol spotting aims to effectively extract and interpret this information in graphic primitive granularity.  Accurately spotting symbols is crucial for  Building Information Modeling(BIM), as the picture shows here: https://anonymous.4open.science/r/x-BB39/bim.png .
>
> ### 2. Clarification on Methodology and Experiments
>
> * **Regarding Equation (2) and Primitive Representation**:  $l_k$ represents the distance between $v_1$ and $v_2$ for linear primitives. For circular primitives like circles and ellipses, $l_k$ is defined as the circumference.  For arcs, $l_k$ means the length of curve, not the distance between $v_1$ and $v_2$. Since all arcs in CAD are quadratic Bezier curves, we can easily compute the length of curves using the python package `svgpathtools`
> * **On Defining Neighborhood $M(p_i)$ in Equation (3):**   In $M(p_i)$, adjacent points means top K nearest primitives which are measured by the distance between the primitive positions (Eq 1) and $p_i$, not locally connected primitives.  $A(p_i)$ means a combination of $M(p_i)$ and $C(p_i)$ which means locally connected primitive points (Eq. 8). The reason that we use a small threshold in Eq. 7 to define the connectivity of two primitives is that for most CAD drawings, the interconnected endpoints of two connected lines are not the same point, but two points that are very close to each other.  We therefore regard lines whose endpoints are close enough as connected.
> * **Motivation Behind KNN Interpolation (Section 3.5):**   This KNN interpolation is used to extract different levels (resolutions) of features in symbol region (i.e, mask region). Although it may lose some information of the symbol in low resolution, but it provides a coarse estimated symbol region at different resolutions, which could be used for pyramid feature extraction as in [R1]. These pyramid of features, including both high-resolution and low-resolution features will be fed into the spotting head for symbol mask/label predicting. Note that we do not only use low-resolution features but different levels of features for final segmentation and label prediction.
> * **Downsampling and Upsampling in Point Transformer:** The downsampling and upsampling processes of point features is shown in https://anonymous.4open.science/r/x-BB39/pool&unpool.png, which is from [R2]. Adjacent point features are downsampled by pooling to get a point feature, which could also be upsampled through unpooling.
> * **Explanation of Equation (12):**  $e_i$ is a graphical entities, $L(e_i)$ is the length of the graphical entitity $e_i$
> * **On Experimental Evaluation:**  (1) Semantic Symbol Segmentation does not distinguish between objects of the same category, while Panoptic Symbol Segmentation does. (2) We follow previous works such as PanCADNet [R3], and GAT-CADNet [R4] to use F1 as metric for fair comparison;  (3) The F1 is the harmonic mean of Precision and Recall, The wF1 is length-weighted F1. Both metrics are well defined in [R3]. We will provide the detail definition in the supplementary material in future.
>
> [R1] Bowen, Cheng, et al. "Masked-attention mask transformer for universal image segmentation" CVPR. 2022.
>
> [R2] Xiaoyang Wu, et al. "Point Transformer V2: Grouped Vector Attention and Partition-based Pooling" CVPR. 2021.
>
> [R3] Zhiwen, Fan, et al. "Floorplancad: A large-scale cad drawing dataset for panoptic symbol spotting" ICCV. 2021.
>
> [R4]Zhaohua, Zheng, et al. "Gat-cadnet: Graph attention network for panoptic symbol spotting in cad drawings" CVPR. 2022.

---

> > ### Comment · Reviewer_q1T8 · 2023-11-22
> >
> > Dear authors, thank you for your detailed response, addressing most of my concerns. I do not require further clarifications. I will carefully review your response along with the other reviewer's comments before making my final recommendation.

---

### Official Review · Reviewer_AtGi · 2023-10-31

**Soundness:** 3 good
**Presentation:** 3 good
**Contribution:** 3 good
**Rating:** 8
**Confidence:** 5

**Summary:**

This paper, titled SymPoint, advocates for representing a symbol as a point and extends the previous methodology to encompass a broader range of symbol properties. The Point Transformer serves as the foundational feature extraction tool. Mask attention and a contrastive connectivity learning mechanism are integrated into the panoptic symbol spotting task, aiming to cultivate rich features that can effectively differentiate between graphic primitives. The PQ performance has been elevated from the previous method to a novel tier, as delineated in the experimental section.

**Strengths:**

With the advancements in point cloud processing and the Transformer architecture, the authors suggest leveraging these powerful backbones from other domains and adapting them to address the challenge of panoptic symbol spotting.
A suite of techniques, encompassing vector graphics representations, Point Transformers, Masked Attention, Contrastive Connection Learning, and KNN Interpolation, has been integrated into the targeted task.
Experimental outcomes reveal that SymPoint significantly outperforms existing methods, exhibiting a considerable advantage in Semantic Symbol Spotting, Instance Symbol Spotting, and Panoptic Symbol Spotting.

**Weaknesses:**

- A primary concern from the reviewer centers on the paper's predominant reliance on existing methodologies to address the issue. Specifically, in Sec3.1 (From Symbol to Point), many parameterizations echo those found in FloorplanCAD, albeit this paper seeks to enhance the diversity of encoded features. The point-based representation in Sec 3.2 directly employs the Point Transformer, reminiscent of CADTransformer. Both Contrastive Connection Learning (Sec3.4) and KNN Interpolation (Sec3.5) have been thoroughly examined in other scholarly works. While the "Attention with Connection Module" presents as novel to the reviewer, it would be beneficial to undertake a comprehensive review to discern if analogous concepts have been previous literature.
- In Table 4, where the benchmark approach registers a PQ of 73.1, could you detail the design of how this baseline method is formulated?
- In Table 4, it appears the newly introduced "ACM" module inadvertently undermines performance. Could the authors shed light on the causative factors behind this decline?
- Again, referencing Table 4, the KInter technique emerges as a salient contributor to performance enhancement. Could the authors offer a more clear explanation and visualization? It might also be worthwhile to highlight this module within the methods section.
- As the proposed framework incorporate a bunch of techniques for a specific application, did you submit the code for reviewing?

**Questions:**

See the raised concens in Weaknesses section

---

> ### Author Response · Authors · 2023-11-16
>
> Dear Reviewer AtGi,
>
> We sincerely appreciate your comments and your detailed questions. We address your questions as follows.
>
> ###  1. Reliance on Existing Methodologies
>
> * While our approach to primitive features bears a resemblance to the vertex features defined in GAT-CADNet, our method focuses on leveraging the most straightforward and fundamental features in CAD drawings, such as type, length, and clockwise angle. These features, while simple, are highly effective in capturing the essence of individual primitives. Although more complex features could potentially be utilized, we found that these basic attributes yield impressive results, striking a balance between simplicity and effectiveness.
> * Compared to CADTransformer, our method is simple yet effective, marked by three key distinctions:
>   * Our process eliminates the need for rasterization of CAD drawings to images, which is a time-intensive step required for many CAD segmentation methods including CADTransformer.
>   * We rely on simple and direct primitive features, as opposed to CADTransformer's use of HRNet for feature extraction. This approach not only simplifies the process but also enhances the scalability of our method to complex, high-resolution CAD drawings.
>   * Our method employs the Point Transformer for interactions with each graphic primitive, whereas CADTransformer utilizes the Vision Transformer (ViT), which more easily occurs OOM when dealing with high resolution CAD drawings;
>   * Our methodology is designed for end-to-end training without the necessity for any post-processing operations. In contrast, CADTransformer requires the use of a clustering algorithm to derive the final instance results, adding additional steps to the process;
>
> ### 2.  Design of Baseline Method
>
> Our baseline is a simple combination of Point Transformer and Mask2Former's transformer decoder with input of point-based representation.  This serves as a foundation for comparing the enhanced performance of SymPoint.
>
> ### 3. Performance of the ACM Module
>
> Our original intention in introducing ACM was to utilize these connections between each graphic primitive.  we conduct experiments in SESYD-floorplans dataset that is smaller than floorplanCAD,  ACM can significantly promote performance and accelerate the model convergence.  An convergence curve between without/with ACM is shown here: https://anonymous.4open.science/r/x-BB39/sesyd_val_loss.png, https://anonymous.4open.science/r/x-BB39/sesyd_val_PQ.png, https://anonymous.4open.science/r/x-BB39/sesyd_val_RQ.png, https://anonymous.4open.science/r/x-BB39/sesyd_val_SQ.png,   The quantitative results are shown below,
>
> |              | PQ    | RQ    | SQ    |
> | ------------ | ----- | ----- | ----- |
> | baseline     | 88.23 | 91.61 | 96.01 |
> | baseline+ACM | 91.98 | 95.99 | 95.85 |
>
> But, floorplanCAD is more complex compared with SESYD-floorplans, and more noisy connections could be introduced. As this figure shows : https://anonymous.4open.science/r/x-BB39/noisy-connection.png . Therefore, we have introduced an additional Contrastive Connection Learning to mitigate the impact of noise connections and more effectively utilize connection information with category consistency.  Ablation study in Table 4 (a) verifies the effectiveness of these modules.
>
> ### 4. Explanation and Visualization of KNN interpolation Technique
>
> While bilinear interpolation, as utilized in Mask2Former, is tailored for regular data, such as image,  but it is  unsuitable for irregular sparse primitive points. Here, We provided some visualizations of point masks for KNN interpolation and bilinear interpolation as shown in https://anonymous.4open.science/r/x-BB39/vis2-knn_interp-vs-bilinear_interp.png.  Note that these point masks are soft masks (ranging from 0 to 1) predicted by intermediate layers. After downsampling the point mask to 4x and 16x, we can clearly find that KNN interpolation well perverse the original mask information interpolation, while  bilinear interpolation causes a significant information loss, which could harm the final performance.
>
> ### 5. Submission of Code for Review
>
> We appreciate the emphasis on the practical application of our framework.  we promise to release our source code for result reproduction and promote the development of this field.

---

> > ### Comment · Reviewer_AtGi · 2023-12-02
> > **Thanks for the detailed explaination and the promise of releasing the code**
> >
> > The reviewer would first like to thank the authors for their responses, including additional experiments, clarifications, and the promise of contributing to the vector recognition community.
> >
> > Reviewer #u21w raises similar concerns to mine, where most of the proposed techniques are existing ones and the authors utilize them to address a new question. However, after reading the rebuttal from the authors, I am convinced that the authors do not simply adopt these techniques for Panoptic Symbol Spotting. Instead, they use existing techniques as baselines and tailor them to address vector data, which includes not only point-based primitives but also various attributes (e.g., type, length, etc.)
> >
> > Furthermore, the proposed techniques demonstrate their generalization to different vector data (user sketches), validating that the proposed techniques are not overfitted to the CAD data domain. Of course, I fully understand that the authors specifically target the problem of Symbol Spotting, and the proposed techniques may not generalize as well to different data domains.
> >
> > Lastly (and perhaps most importantly), the promise of contributing to the community by submitting the code in the near future will be very helpful for the reproduction of your results and to validate that your method is open, general, and strong. A schedule for the timeline of the submission would further **solidify** the work.
> >
> > Considering all factors above, I am inclined to raise my score to ACCEPT.

---

> > > ### Comment · Reviewer_AtGi · 2023-12-02
> > > **Additional commments**
> > >
> > > To ensure accessible validation of your method, please adequately prepare your open-source materials, which should be able to reproduce all results in the main draft and the uploaded documents (https://anonymous.4open.science/r/x-BB39/).

---

### Official Review · Reviewer_u21w · 2023-11-01

**Soundness:** 2 fair
**Presentation:** 2 fair
**Contribution:** 2 fair
**Rating:** 3
**Confidence:** 4

**Summary:**

In this paper, a method for symbol spotting from CAD vector graphics (VG), called SymPoint, is proposed. SymPoint treats graphic primitives as a set of 2D points. Two strategies, attention with connection module (ACM) and contrastive connection learning (CCL), are devised to better utilize the local connection information of primitives and enhance their discriminability.

**Strengths:**

The main idea and technical detailed are clearly presented.

**Weaknesses:**

1. The originality and technical contribution of this work is quite limited. Point Transformer, Mask2Former and InfoNCE are all well-established methods or models.
2. The potential application range of the proposed method can be narrow (CAD vector graphics), because it is unclear whether the idea and techniques presented in this work can be extend ed to other tasks.

**Questions:**

The authors should explain and verify the originality and technical contribution of the proposed method.

---

> ### Author Response · Authors · 2023-11-16
>
> Dear Reviewer u21w,
>
> We sincerely appreciate your comments and your detailed questions. We address your concerns as follows.
>
> **1. Regarding Originality and Technical Contribution:**
>
> While SymPoint incorporates well-established methods like Point Transformer, Mask2Former, and InfoNCE, our innovation lies in adapting and optimizing these methods for the specific challenge of CAD vector graphics recognition.  **It is worth noting that the combination of Point Transformer and Mask2Former is just our baseline.**
>
> Specifically, the novelties of our paper lie in several aspects:
>
> * We carefully analyzing the data characteristics of CAD drawings and design novel and effective way of transferring CAD primitive entities into point feature in Section 3.1;
> * We propose Attention with Connection Module (ACM) , which is novel in efficiently capturing the local connection information of primitives and enhancing their discriminability when in conjunction with Contrastive Connection Learning (CCL) module in the context of CAD graphics in Section 3.3 and 3.4 ;
> * We propose KNN Interpolation to downsample  attention mask for masked attention calculation, which shows superior performance compared with the naive bilinear interpolation used in Mask2Former in Section 3.5.
>
> Our ablation study in Table 4 (a) has demonstrated that these modules effectively promoted the performance of the model. These strategies represent a unique approach not previously explored in this field.  **We achieves 77.3/87.1/ 88.7 (PQ/RQ/SQ) with these novel designs, significantly outperforming the baseline 73.1/83.3/87.7 (PQ/RQ/SQ) . Note that our baseline method has already surpassed many state-of-the-art methods, such as PanCADNet and CADTransformer.**
>
> **2. On the Potential Application Range:**
>
> In response to the concern regarding the application range of SymPoint, we acknowledge that our current focus is primarily on CAD vector graphics. However, we believe there is potential for applying our method to a broader range of fields, like **3D modeling[R1]** and **circuit design[R2]**, **sketch segmentation[R3]**  as shown here:  https://anonymous.4open.science/r/x-BB39/application.png. **We have conduct some preliminary experiments on circuit design in the appendix A.2. and achieve state-of-the-art performance on two datasets as shown in Table 7.** Although this beyond the scope of this paper, we see it as a valuable direction for future research.
>
> [R1] Lv, Xiaolei, et al. "Residential floor plan recognition and reconstruction" ICCV. 2021.
>
> [R2] Jiang, Xinyang, et al. "Recognizing vector graphics without rasterization" NeurIPS. 2021.
>
> [R3]Yang, Lumin , et al. "Sketchgnn: Semantic sketch segmentation with graph neural networks" TOG. 2023.

---

### Author Response · Authors · 2023-11-20
**Thank you and expect more disucssions**

Dear My Reviewer u21w, AtGi, q1T8,

Thank you for your support and helpful comments. We've tried our best to address your concerns, and we hope our responses make sense to you. Importantly, we much value your comments and would be happy to discuss more. **If you have any additional questions or open discussions, please don't be hesitant to leave more comments. We are always available at all time, to actively address any concerns or be prepared for more discussions**.

**Your opinions are rather important for us to improve the work!**

Thank you!

Sincerely,

Authors

---

### Meta-Review · Area_Chair_ZwAi · 2023-12-12

**Metareview:**

The paper has received two acceptance and one rejection recommendations with the rating 3, 6, 8. The major concern raised by reviewer u21w is that the proposed method is not novel because the basic building blocks such as Point Transformer, Mask2Former and InfoNCE are well known methods. After considering the rebuttal, reviewer AtGi champions the paper and is convinced that the paper assembles related techniques and turns them into a novel algorithm effectively solving the problem in symbol spotting. After carefully considering the review and rebuttal, the AC concurred with reviewer AtGi that the paper proposes a novel algorithm. Overall, the AC does not find enough evidence to reject the paper and concurs with the acceptance recommendations from reviewer AtGi and q1T8.

**Justification For Why Not Higher Score:**

While the paper writes a nice system paper that adopts several suitable tools for solving the particular Panoptic Symbol Spotting problem, the paper shows less components that would inspire general audience for problems in other domains. Therefore, the AC recommends accepting the paper with poster.

**Justification For Why Not Lower Score:**

The paper is a solid system paper with an interesting application. The AC does not find the novelty issue raised by reviewer u21w and concurs with the assessments from other reviewers.

---

### Decision · Program_Chairs · 2024-01-16

Accept (poster)